# LOCAL INFORMATION OPPONENT MODELLING USING VARIATIONAL AUTOENCODERS

## ABSTRACT

Modelling the behaviours of other agents (opponents) is essential for understanding how agents interact and making effective decisions. Existing methods for opponent modelling commonly assume knowledge of the local observations and chosen actions of the modelled opponents, which can significantly limit their applicability. We propose a new modelling technique based on variational autoencoders, which are trained to reconstruct the local actions and observations of the opponent based on embeddings which depend only on the local observations of the modelling agent (its observed world state, chosen actions, and received rewards). The embeddings are used to augment the modelling agent's decision policy which is trained via deep reinforcement learning; thus the policy does not require access to opponent observations. We provide a comprehensive evaluation and ablation study in diverse multi-agent tasks, showing that our method achieves comparable performance to an ideal baseline which has full access to opponent's information, and significantly higher returns than a baseline method which does not use the learned embeddings.

## 1 INTRODUCTION

An important aspect of autonomous decision-making agents is the ability to reason about the unknown intentions and behaviours of other agents. Much research has been devoted to this *opponent modelling* problem [2], with recent works focused on the use of deep learning architectures for opponent modelling and reinforcement learning (RL) [20, 34, 16, 33].

A common assumption in existing methods is that the modelling agent has access to the local trajectory of the modelled agents [2], which may include their local observations of the environment state, their past actions, and possibly their received rewards. While it is certainly desirable to be able to observe an agent's local context in order to reason about its past and future decisions, in practice such an assumption may be too restrictive. Agents may only have a limited view of their surroundings, communication with other agents may not be feasible or reliable [40], and knowledge of the perception system of other agents may not be available [13]. In such cases, an agent must reason with only locally available information.

We consider the question: *Can effective opponent modelling be achieved using only the locally available information of the modelling agent during execution?* A strength of deep learning techniques is their ability to identify informative features in data. Here, we use deep learning techniques to extract informative features from a stream of local observations for the purpose of opponent modelling.

Specifically, we consider multi-agent settings in which we control a single agent which must learn to interact with a set of opponent agents (we use the term "opponent" in a neutral sense). We assume a given set of possible policies for opponent agents and that these policies are fixed (that is, other agents do not simultaneously learn, such as in multi-agent RL [32]). We propose an opponent modelling method which is able to extract a compact yet informative representation of opponents given only the local information of the controlled agent, which includes its local state observations, past actions, and rewards. To this end, we use an encoder-decoder architecture based on variational autoencoders (VAE) [26]. The VAE model is trained to replicate opponent actions and observations from the local information only. During training, the opponent's observations are utilised as reconstruction targets for the decoder; after training, only the encoder component is retained which generates embeddings using local observations of the controlled agent. The learned embeddings condition the policy of the

controlled agent in addition to its local observation, and the policy and VAE model are optimised concurrently during the RL learning process.

We evaluate our proposed method, called *Local Information Opponent Modelling* (LIOM), in two benchmark environments used in multi-agent systems research, the multi-agent particle environment [31, 28] and level-based foraging (LBF) [1]. Our results support the idea that effective opponent modelling can be achieved using only local information during execution: the same RL algorithm generally achieved higher average returns when combined with our opponent embeddings than without, and in some cases the average returns are comparable to those achieved by an ideal baseline which has full access to the opponent's trajectory. We evaluate the method's ability to predict the opponent's actions, and provide an ablation study on the different types of local information used by the encoder.

## 2  RELATED WORK

**Learning Opponent Models:**  We are interested in opponent modelling methods that use neural networks to learn representations of the opponents. He et al. [20] proposed a method which learns a modelling network to reconstruct an opponent's actions given the opponent's observations. Raileanu et al. [34] developed an algorithm for learning to infer an opponents' intentions using the policy of the controlled agent. Grover et al. [16] proposed an encoder-decoder method for modelling the opponent's policy. The encoder learns a point-based representation of different opponent trajectories, and the decoder learns to reconstruct the opponent's policy. In addition, the authors introduced an objective to separate embeddings of different agents into different clusters. Rabinowitz et al. [33] proposed the Theory of mind Network (TomNet), which learns embedding-based representations of opponents for meta-learning. Tacchetti et al. [42] proposed relational forward models to model opponents using graph neural networks. A common assumption in these methods, which our work aims to eliminate, is that the modelling agent has full access to the opponent's local information during execution, including their observations, chosen actions, and received rewards.

Opponent modelling from local information has been researched under the I-POMDP model [13] and in the Poker domain research. In contrast to our work, I-POMDPs utilise recursive reasoning [2] which assumes knowledge of the observation models of the modelled agents (which is not available in our setting). In the Poker domain, Johanson et al. [24] proposed Restricted Nash Response (RNR) for computing robust counter-strategies to opponents. Additionally, they generate a mixture-of-experts counter-strategies to various opponents. During execution, the UCB1 algorithm [4] is used to adapt and select the appropriate counter-strategy out of the mixture against each specific and previously unknown opponent. Bowling et al. [7] propose a method for online evaluation of an agent's strategy using importance sampling for reducing the variance of the estimation. Bard et al. [5] combined several ideas from the aforementioned works to build a complete Poker agent system. Their method creates mixture-of-experts strategies, and during execution they deploy Exp4 [3] for online adaptation (selection of the best strategy from the mixture) to each opponent. The aforementioned works do not require any access to the opponent's observations during execution. The main difference between our method and these works is that the latter require a number of online adaptation episodes (to select the best strategy) against each opponent. In contrast, in our work we use a single episode for adaptation.

**Representation Learning in Reinforcement Learning:**  Another related topic which has received significant attention is representation learning in RL. Using unsupervised learning techniques to learn low-dimensional representations of the environment state has led to significant improvements in RL. Ha and Schmidhuber [18] proposed a VAE-based model and a forward model to learn state representations of the environment. Hausman et al. [19] learned task embeddings and interpolated them to solve more difficult tasks. Igl et al. [22] used a VAE model for learning state representations in partially-observable environments. Gupta et al. [17] proposed a model which learns Gaussian embeddings to represent different tasks during meta-training and manages to quickly adapt to new task during meta-testing. Gregor et al. [14] developed a VAE-based model for long-term state predictions. The work of Zintgraf et al. [44] is closely related, where the authors proposed a recurrent VAE model which receives as input the observation, action, reward of the controlled agent and learns a variational distribution of tasks. Rakelly et al. [35] used representations from an encoder for off-policy meta-RL. Note that all of these methods were designed for learning representations of tasks or properties of the environment. In contrast, our approach focuses on learning representations of opponents.

**Multi-agent Reinforcement Learning (MARL):** This term is used to describe the learning proce-dure of multiple agents in a shared multi-agent environment. At the beginning of the training, the agents are usually untrained and initialised with a random policy. Using an opponent's trajectory during training but not during execution, is a common MARL paradigm, called Centralised Training Decentralised Execution (CTDE). The opponent's information can be utilised during training for various reasons, such as computing a joint value function [28, 9, 41, 36], or generating intrinsic rewards [23]. However, during execution only the local information of each agent is used for selecting actions in an environment. While our work draws connection with CTDE, we consider a different problem than the one that MARL addresses. In this work, we have a set of fixed opponent policies, and we train a *single agent* to model and interact with the policies in this set, without accessing the opponent's trajectory during execution.

## 3 TECHNICAL PRELIMINARIES

### 3.1 REINFORCEMENT LEARNING

We model the decision problem as a Markov Decision Processes (MDP). An MDP consists of a set of states $\mathcal{S}$, a set of actions $\mathcal{A}$, a transition function, $P(s'|s,a)$, specifying the probability of the next state, $s'$, after taking action $a$ in state $s$, and a reward function, $r(s',a,s)$, which returns a scalar value conditioned on two consecutive states and the intermediate action. A policy function is used to choose an action in given a state, which we assume can be stochastic, $a \sim \pi(a|s)$. Given a policy $\pi$, the state-value function is defined as $V^\pi(s_t) = \mathbb{E}_\pi[\sum_{i=t}^H \gamma^{i-t} r_t | s = s_t]$ and the action-value (Q-value) function $Q^\pi(s_t, a_t) = \mathbb{E}_\pi[\sum_{i=t}^H \gamma^{i-t} r_t | s = s_t, a = a_t]$, where $0 \leq \gamma \leq 1$ is the discount factor and $H$ is the finite horizon of the episode. RL methods aim to compute an optimal policy that maximises the state value functions. In this work, we will use an algorithm called Synchronous Advantage Actor-Critic (A2C) [30, 8]. A2C is an on-policy actor-critic algorithm which uses parallel environment copies to break the correlation between consecutive experience samples. Assuming a policy $\pi_\theta$ and value network $V_\phi$ with parameters $\theta$ and $\phi$, respectively, the actor-critic parameters are optimised via

$$\min_{\theta,\phi} \mathbb{E}_B[-\hat{A}\log\pi_\theta(a|s) + \frac{1}{2}(r + \gamma V_\phi(s') - V_\phi(s))^2] \tag{1}$$

where $B$ is the batch of transitions, $B = \{(s_t, a_t, r_t, s_{t+1})\}_{t=0}^{t=|B|}$, and $\hat{A}_t = r_t + \gamma V_\phi(s_{t+1}) - V_\phi(s_t)$ is the basic advantage term (in experiments we use Generalised Advantage Estimation (GAE) [38]; see Appendix D).

### 3.2 VARIATIONAL AUTOENCODER

Consider samples from a dataset $x \in \mathcal{X}$ which are generated from some hidden (latent) random variable $z$ based on a generative distribution $p_{\boldsymbol{u}}(x|z)$ with unknown parameter $\boldsymbol{u}$, and a prior distri-bution on the latent variables which we assume to be Gaussian with zero mean and unit variance, $p(z) = \mathcal{N}(z; \boldsymbol{0}, \boldsymbol{I})$. We seek to approximate the true posterior $p(z|x)$ with a variational parametric distribution $q_{\boldsymbol{w}}(z|x) = \mathcal{N}(z; \boldsymbol{\mu}, \boldsymbol{\Sigma}, \boldsymbol{w})$. Kingma and Welling [26] proposed the Variational Autoen-coder (VAE) to learn this distribution. Starting with the Kullback-Leibler (KL) divergence from the approximate to the true posterior, $D_{\mathrm{KL}}(q_{\boldsymbol{w}}(z|x)\|p(z|x))$, the lower bound on the evidence $\log p(x)$ (ELBO) is derived as:

$$\log p(x) \geq \mathbb{E}_{z \sim q_{\boldsymbol{w}}(z|x)}[\log p_{\boldsymbol{u}}(x|z)] - D_{\mathrm{KL}}(q_{\boldsymbol{w}}(z|x)\|p(z)) \tag{2}$$

Maximising the ELBO leads to minimisation of the KL divergence from the approximate to the true posterior. The architecture consists of two neural networks: the encoder which receives a sample $x$ and generates the Gaussian variational distribution $q(z|x; \boldsymbol{w})$; and the decoder which receives a sample from the Gaussian variational distribution and reconstructs the generative distribution $p_{\boldsymbol{u}}(x|z)$. The architecture is trained using the reparameterisation trick [26]. Zhao et al. [43] noticed that the KL divergence can lead to an uninformative posterior and proposed the use of the Maximum Mean Discrepancy (MMD) [6, 15] for forcing the posterior to be close to the prior.

$$D_{\mathrm{MMD}}(q(z)\|p(z)) = \mathbb{E}_{z,z'\sim q}[k(z,z')] + \mathbb{E}_{z,z'\sim p}[k(z,z')] - 2\mathbb{E}_{z\sim q,z'\sim p}[k(z,z')] \tag{3}$$

where $k(z,z')$ is a Gaussian kernel.

## 4 APPROACH

### 4.1 PROBLEM FORMULATION

We control a single agent which must learn to interact with other agents (opponents) that use one of a fixed number of policies. We model this as a Markov game [27] which consists of $N$ agents $\mathbb{I} = \{1, 2, ..., N\}$, a state space $\mathcal{S}$, the joint action space $\mathcal{A} = \mathcal{A}_1 \times ... \times \mathcal{A}_N$, a transition function $P : \mathcal{S} \times \mathcal{A} \times \mathcal{S} \to [0, 1]$ specifying the transition probabilities between states given a joint action, and for each agent $i \in \mathbb{I}$ a reward function $r_i : \mathcal{S} \times \mathcal{A} \times \mathcal{S} \to \mathbb{R}$. We consider partially-observable settings, where each agent $i$ has access only to its local observation $o_i \subset s \in \mathcal{S}$ and reward $r_i$. We denote the agent under our control by 1, and the opponent agents by $-1$ where for notational convenience we will treat the opponent agents as a single "combined" agent with joint observations $o_{-1}$ and actions $a_{-1}$. We assume a set of opponent policies, $\Pi_{-1} = \{\pi_{-1}^k | k = 1, ..., K\}$, which may be defined manually (heuristic) or pretrained using RL. Each opponent policy determines the opponent agent's actions as a mapping $\pi_{-1}^k(o_{-1})$ from the opponent's local observation $o_{-1}$ to a distribution over actions $a_{-1}$. Our goal is to find a policy $\pi_\theta$ parameterised by $\theta$ for agent 1 which maximises the average return against opponents from the training set $\Pi_{-1}$, assuming that each opponent policy is initially equally probable and fixed during an episode:

$$\arg \max_\theta \mathbb{E}_{\pi_\theta, \pi_{-1} \sim \mathcal{U}(\Pi_{-1})} \left[ \sum_{t=1}^{H} \gamma^t r_{1,t} \right] \quad (4)$$

where $r_{1,t}$ is the reward received by agent 1 at time $t$, $H$ is the episode length (horizon), and $\gamma \in (0, 1)$ is a discount factor.

### 4.2 LOCAL INFORMATION VARIATIONAL AUTOENCODERS

We denote by $\tau_{-1} = \{o_{-1,t}, a_{-1,t}\}_{t=0}^{t=H}$ an opponent trajectory where $o_{-1,t}$ and $a_{-1,t}$ are the opponent's observation and action at time step $t$ in the trajectory, up to horizon $H$. These trajectories are generated from the opponent policies in $\Pi_{-1}$, which are represented in a latent (or *embedding*) space $\mathcal{Z}$. Additionally, we assume that there exists an unknown generative model $p_{\boldsymbol{u}}(\tau_{-1}|z), z \in \mathcal{Z}$. The latent variable $z$ contains information about the trajectory of the opponent. Our approach is to solve the problem defined in Section 4.1 by performing reinforcement learning in the joint space of the observation space of our agent and the latent space of the opponent. We aim to approximate the unknown posterior, $p(z|\tau_{-1})$, using a variational Gaussian distribution $\mathcal{N}(\boldsymbol{\mu_w}, \boldsymbol{\Sigma_w})$ with parameters $\boldsymbol{w}$. As a result, during execution, we can sample the latent variable from the approximate posterior $z \sim \mathcal{N}(z; \boldsymbol{\mu_w}, \boldsymbol{\Sigma_w})$.

In Sections 1 and 2, it was noted that most agent modelling methods assume access to the opponent's observations and actions both during training and execution. To remove this assumption during execution, we propose a VAE that uses a parametric variational distribution which is conditioned on the observation-action-reward triplet of the agent under our control; $q_{\boldsymbol{w}}(z|\tau_{1,:t} = (o_{1,1:t}, a_{1,1:t-1}, r_{1,1:t-1}))$. Specifically, we approximate the true posterior that is conditioned on opponent's information, with a variational distribution that only depends on local information. We start by writing the KL divergence from the approximate to the true posterior:

$$D_{\text{KL}}(q_{\boldsymbol{w}}(z|\tau_1) \| p(z|\tau_{-1})) \quad (5)$$

By following the works of Kingma and Welling [26] and Zhao et al. [43], the loss can be written as:

$$\mathcal{L}(\tau_1, \tau_{-1}; \boldsymbol{w}, \boldsymbol{u}) = - \mathbb{E}_{z \sim q_{\boldsymbol{w}}(z|\tau_1)} \left[ \log p_{\boldsymbol{u}}(\tau_{-1}|z) \right] + \lambda D_{\text{MMD}}(q_{\boldsymbol{w}}(z|\tau_1) \| p(z)) \quad (6)$$

From Equation 6, we observe that the variational distribution depends only on locally available information. Since during execution only the encoder is required to generate the opponent's model, this approach removes the assumption that access to the opponent's observations and actions is available during execution. At each time step $t$, the recurrent encoder network generates a latent sample $z_t$, which is conditioned on the information of the agent under control $(o_{1,1:t}, a_{1,1:t-1}, r_{1,1:t-1})$, until time step $t$.

The first term of the VAE loss consists of the reconstruction loss of the opponent's trajectory which involves the observations and actions of the opponent. The opponent's observation depends on the

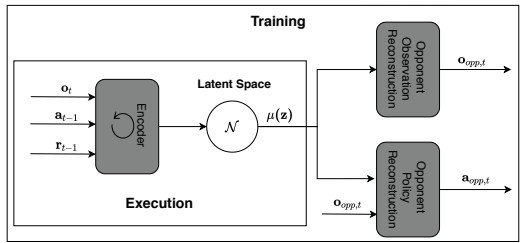

Figure 1: Diagram of LIOM architecture.

dynamics of the environment. The opponent's action at each time step depends on its observation and the opponent's identity, which is represented by the latent variable $z$. Therefore, the decoder consists of two fully-connected feed-forward networks.

$$\log p_{\boldsymbol{u}}(\tau_{-1}|z) = \sum_{t=1}^{H} \log p_{\boldsymbol{u}}(o_{-1,t}, a_{-1,t}|z_t) = \sum_{t=1}^{H}[\log p_{\boldsymbol{u}}(o_{-1,t}|z_t) + \log p_{\boldsymbol{u}}(a_{-1,t}|o_{-1,t}, z_t)] \quad (7)$$

From Equation 7, we observe that conditioned on the latent variable, the decoder reconstructs the opponent's observation and the opponent's action given the opponent's observation. Intuitively, $z_t$ encodes the type of policy used by the opponent at the current episode and its observation at time step $t$. Figure 1 illustrates the proposed VAE as well as the information that requires during training and execution. We refer to this method as **LIOM** (**L**ocal **I**nformation **O**pponent **M**odelling). LIOM uses the information of both the controlled agent and opponent during training, but during execution only the information of the controlled agent is used.

### 4.3 Reinforcement Learning Training

We use the latent variable $z$ augmented with our agent's observation to condition the policy of our agent, which is optimised using RL. Consider the augmented observation space $\mathcal{O}' = \mathcal{O} \times \mathcal{Z}$, where $\mathcal{O}$ is the original observation space of our agent in the Markov game, and $\mathcal{Z}$ is the representation space of the opponent models. The advantage of learning the policy on $\mathcal{O}'$ compared to $\mathcal{O}$ is that the policy can adapt to different $z \in \mathcal{Z}$. We optimise the policy of the controlled agent, using the A2C algorithm (cf. Sec. 3.1). Note that other RL algorithms could be used in place of A2C. The input to the actor and critic are the local observation and the mean of the variational distribution. We do not back-propagate the gradient from the actor-critic loss (Equation 1) to the parameters of the encoder. We use different learning rates for optimising the parameters of the networks of A2C and LIOM. LIOM is a VAE model, and we empirically observed that it exhibits high stability during learning, allowing us to use larger learning rate compared to RL. Additionally, we subtract the policy entropy from the policy gradient loss to encourage exploration [30]. Given a batch $B$ of collected trajectories in the environment, the update equation for our proposed method is the following:

$$\min_{\boldsymbol{\phi},\boldsymbol{\theta}} \mathbb{E}_B\Big[\frac{1}{2}\big(r_1 + \gamma V_{\boldsymbol{\phi}}(o_1', \mu(z')) - V_{\boldsymbol{\phi}}(o_1, \mu(z))\big)^2 - \hat{A}\log\pi_{\theta}(a_1|o_1, \mu(z)) - \beta H(\pi_{\theta}(a_1|o_1, \mu(z)))\Big] \quad (8)$$

$$\min_{\boldsymbol{w},\boldsymbol{u}} \mathbb{E}_B\Big[-\log p_{\boldsymbol{u}}(o_{-1}|z) - \log p_{\boldsymbol{u}}(a_{-1}|o_{-1}, z) + \lambda D_{\text{MMD}}(q_{\boldsymbol{w}}(z|\tau_1)\|p(z))\Big] \quad (9)$$

The pseudocode of LIOM is given in Appendix A. Intuitively, at the beginning of each episode, LIOM starts with an uninformative posterior which is equal to the prior (isotropic Gaussian) over the possible opponents. At each time step, the agent interacts with the environment and the opponent and updates the posterior over opponents based on the local information that it perceives.

## 5 Experiments

We evaluate LIOM in several multi-agent tasks and compare the average returns during RL training against two baselines. We evaluate the embeddings learned by LIOM's encoder and the reconstruction accuracy of the decoder. Finally, we test the stability of LIOM with respect to different inputs in the encoder and the effect of the number of opponent policies in the returns of LIOM.

### 5.1 MULTI-AGENT ENVIRONMENTS

**Speaker-listener:** the environment consists of two agents, called *speaker* and *listener*, as well as three designated landmarks. At the start of each episode, the listener and landmarks are generated in a random position, and are randomly assigned one of three possible colours - red, green, or blue. The task of the listener is to navigate to the landmark that has the same colour as the listener. However, the colour of the listener can only be observed by the speaker, thus the speaker has to learn to communicate the correct colour to the listener. The speaker observes the colour of the goal landmark as a one-hot encoded vector and outputs a 5-bit binary communication message as an action. The listener observes the relative positions of all landmarks and the communicated message of the previous time step and can choose to navigate using the actions forward, backward, right, left, no-op. The speaker observes only the colour of the listener. The reward at each time step is the negative Euclidean distance between the listener and the correct landmark.

**Double speaker-listener:** the environment consists of two agents and three designated landmarks, similarly to the speaker-listener environment. The only difference is that both agents are simultaneously speakers and listeners. Therefore, at the beginning of the episode, each agent has a colour that can only be observed by the other agent. Each agent must learn both to communicate a message to the other agent as well as navigate to the correct landmark. The agent's observation includes the relative positions of all landmarks and the other agent as well as the communication message from the previous timestep and the colour of the other agent. Each agent performs both actions from the previous environment; it communicates the opponent's goal landmark and navigates to its own. The reward at each time step is the negative average Euclidean distance between each agent and the corresponding correct landmark.

**Level-based foraging (LBF):** the environment is a $20 \times 20$ grid-world, consisting of two agents and four food locations. The agents and the foods are assigned random levels and positions at the beginning of an episode. The goal is for the agents to collect all foods. Agents can either move in one of the four directions or attempt to pick up a food. A group of one or more agents successfully pick a food if the agents are positioned in the adjacent cells to the food and if the sum of the agents' levels is at least as high as the food's level. The controlled agent has to learn to cooperate to load foods with a high level and at the same time act greedily for foods that have lower levels. The environment has sparse rewards, representing the contribution of the agent in the gathering all foods in the environment. For example, if the agent receives a food with level 2, and there are another three foods with levels 1, 2 and 3 respectively, the reward of the agent is $2/(1 + 2 + 2 + 3) = 0.25$. Thus, the maximum cumulative reward that both agents can achieve is normalised to 1. The environment is partially-observable, where the agent observes up to four grid cells in every direction, and as a results, it can only perceive foods and the opponent that are within this distance. The agent's observation includes its position and level as well as the relative position and level of each other agent and food in the visible grid cells.

Images of the multi-agent environments are provided in Appendix B. For each environment, we create ten different opponent policies which are used for training (set $\Pi_{-1}$). In speaker-listener we control the listener, and we create ten policies for the speaker using different communication messages for different colours. In double speaker-listener, we create a diverse set of opponent policies that use different communication messages similar to speaker-listener, while they learn to navigate using the MADDPG algorithm [28]. The generated policies are different because they learn to both interpret and communicate different messages to the other agent. For LBF, we created four heuristic policies such as moving to the closest food or closest level-compatible food, and additionally six policies using a stochastic-based RL method with different initial seeds leading to different trained policies. More details about the process of generating opponent policies are presented in Appendix C.

### 5.2 EPISODIC RETURNS DURING TRAINING

**Baselines:** We compare LIOM against two baselines, which are indicative of the upper and the lower performance of LIOM when it is evaluated against the opponent policies from $\Pi_{-1}$. For the upper baseline, we propose a VAE-based opponent model which is trained on the trajectories of the opponents. The encoder of the VAE receives as input the observation and the action of the opponent and infers the opponent model $z$. We call this baseline FIOM (Full Information Opponent Model). FIOM approximates the latent opponent identity distribution using a variational distribution that is

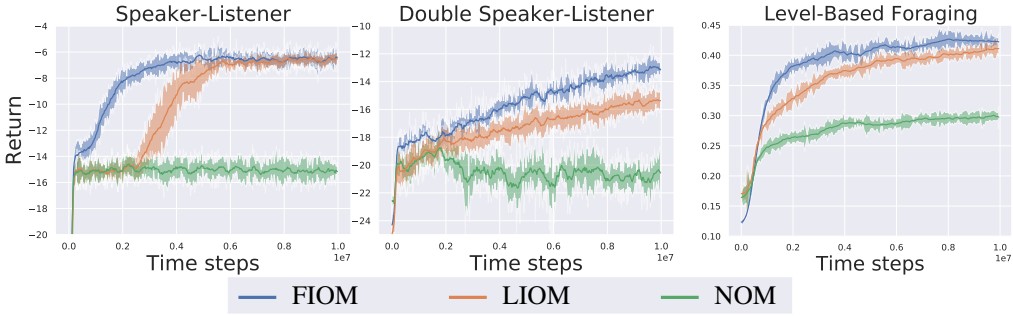

Figure 2: Episodic return and $95\%$ confidence interval against opponent policies from $\Pi_{-1}$.

conditioned on the trajectory of the opponent. We optimise the opponent representation along with the reinforcement learning objective similarly to LIOM. Note that FIOM has access to opponent's information both during training and execution. The lower baseline that we use is called No Opponent Model (NOM). NOM uses the A2C algorithm without any model of the opponent.

Figure 2 shows the episodic returns for the three methods in all three environments. The lines shown in our results plots show the average return over five runs with different initial seeds, and the shadowed part represents the $95\%$ confidence interval. We evaluate the methods every 1000 training episodes for 100 episodes. During the evaluation, we compute the mean of the variational distribution at each time step, and the agent follows the stochastic policy. We found that sampling the action from the policy distribution leads to significantly higher returns compared to following the greedy policy. We observe that LIOM's episodic returns are closer to FIOM than NOM in all environments. This shows that LIOM can successfully learn opponent models using only locally available information. At the beginning of the training the returns achieved by LIOM and NOM are identical, because the encoder generates uninformative and noisy embeddings. After some time step, the encoder has learned to generate informative embeddings and the difference in returns increases significantly.

## 5.3 ENCODER EVALUATION

We analyse the embeddings learned by LIOM's encoder. Figure 3 presents the first two principal components of the mean of the variational distribution at the 20th time step of the episode for all opponents in $\Pi_{-1}$. In speaker-listener and double speaker-listener, we observe that three different clusters are created, representing the three different colours that the opponent observes. We observe that the controlled agent learns to perfectly identify the underlying colour, even though this information exists only in the hidden opponent's observation. Note that the embeddings are not clustered based on the opponent identities but based on the selected actions and private observations, which means that different opponent policies may be embedded in the same region.

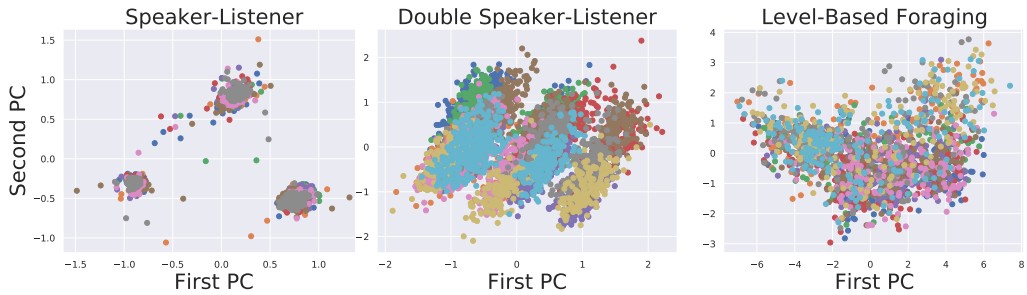

Figure 3: First and second principal components of the learned opponent representations. Points represent individual episodes, colours represent the different opponent policies in $\Pi_{-1}$.

Another interesting question that we aim to answer is how quickly the embeddings converge during an episode. At the beginning of the episode, the model does not know anything about the opponent, and it outputs an uninformative posterior. As the agent interacts with the opponent, the posterior is updated to accommodate the information that has been gathered. To measure the speed of convergence, Figure 4 presents the Euclidean distance between the mean of the variational Gaussian at the each time step, and the mean of the variational Gaussian at the 25th timestep. From Figure 4, we observe that the distance to the position of the last embedding decreases through time in all tasks.

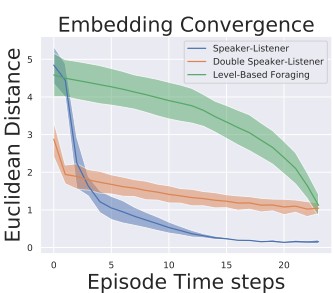

Figure 4: Euclidean distance of embeddings to the embedding at 25th time step (with 95% confidence interval).

## 5.4 DECODER EVALUATION

The decoder receives as input an embedding from the latent space, and predicts the opponent's observation and action. We first evaluate the decoder based on action prediction accuracy. At the 20th time step of the episode, we sample an embedding and use the decoder to reconstruct the opponent's action. Figure 5 presents the average action prediction accuracy (the percentage of the reconstructed actions that match those that were performed from the opponents) and the 95% confidence interval.

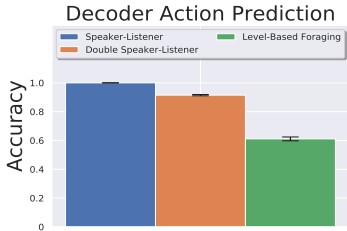

Figure 5: Action prediction accuracy.

| Environment | Decoder | Opponent |
|---|---|---|
| Speaker-listener | $-6.6 \pm 0.6$ | $-6.3 \pm 0.2$ |
| Double speaker-listener | $-14.6 \pm 1.4$ | $-14.4 \pm 1$ |
| Level-based foraging | $0.27 \pm 0.04$ | $0.34 \pm 0.02$ |

Table 1: Average episodic return and 95% confidence interval of the opponent using the decoder vs. opponent's policy to select the opponent's actions.

The difference in prediction accuracy between LBF and the other tasks is explained by the fact that the RL-based opponent policies in LBF are stochastic, and as a result harder to predict. Next, since the decoder is trained to imitate the opponent's policies, we can replace the opponent policies with the decoder and compare the returns achieved by the opponent, shown in Table 1. The performance of the opponent using the decoder is not significantly altered, demonstrating that the decoder learns to imitate the opponent policy. The results of Figure 5 and Table 1 are averaged over 5000 episodes for five different seeds.

## 5.5 ABLATION STUDY ON LIOM INPUTS

Our full method utilises the observation, action, and reward sequence of the controlled agent to generate the opponent model. To evaluate the impact of different types of input data, we use different combinations of inputs in the encoder and compare the episodic returns. Figure 6 presents the average episode return for four different cases: LIOM (full), LIOM using only observations and actions, LIOM using only observations and rewards, LIOM using only actions and rewards. Figure 6 shows that in speaker-listener the most important components are the reward and the observation. In double speaker-listener the performance is affected by the absence of either the agent's actions or observations in the computation of the embeddings. The reward is the average over the distance of both agents to the correct landmarks, which is complex and cannot be effectively utilised by the encoder. In LBF, the average episodic returns are only affected by the absence of the observation in LIOM's input. In LBF, the rewards are sparse and the actions that our agent performs can be inferred from consecutive observations, and absence of any of those terms does not affect the returns. Generally, our experiments indicate that LIOM is robust with respect to different inputs the encoder.

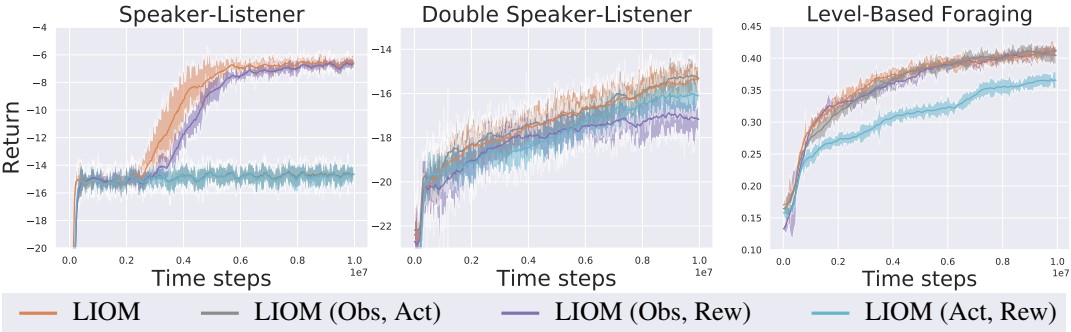

Figure 6: Episodic return and $95\%$ confidence interval against opponent policies from $\Pi_{-1}$ for different combinations of input data for the encoder. (In speaker-listener, LIOM (Obs, Act) overlaps with LIOM (Act, Rew).)

## 5.6 NUMBER OF OPPONENT POLICIES

Finally, we evaluate the effect of number of the opponent policies in the average achieved returns of FIOM, LIOM, and NOM in the double speaker-listener environment. We train FIOM, LIOM, and NOM against a subset $\Pi'_{-1}$ of the original $\Pi_{-1}$ set, where the size of $\Pi'_{-1}$ varies between one and ten. Figure 7 presents the average episodic return achieved at the end of training against different numbers of opponent policies. We observe that when there is a single opponent policy the performance of FIOM, LIOM, and NOM is equal. This is expected as there is not need for opponent modelling. As we increase the size of $\Pi'_{-1}$, we observe a steep decrease in the returns of NOM, while the returns of FIOM, and LIOM decrease at a much slower rate. This is a natural consequence of our problem formulation, since at the beginning of the episode LIOM does not have information about the opponent and because both the controlled agent and the opponent are relatively far

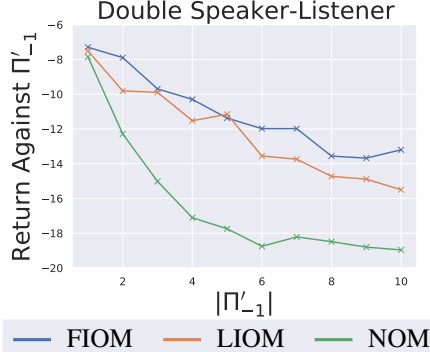

Figure 7: Average episodic return against different numbers of opponent policies.

from the correct landmarks, they are heavily penalised. After some interaction time steps LIOM is able to identify the opponent, and outputs efficient embeddings that can be utilised by the policy network. On the other hand, NOM learns an "average" policy against the opponent policies, resulting in much lower returns.

## 6 CONCLUSION

We proposed a new opponent modelling approach, LIOM, which jointly trains a VAE-based opponent model with a decision policy for the agent under control, such that the resulting opponent model is conditioned only on the local observations of the controlled agent. LIOM is agnostic to the type of interactions in the environment (cooperative, competitive, mixed) and can model an arbitrary number of opponent policies simultaneously in the set $\Pi_{-1}$. Our results show that LIOM can significantly improve the episodic return that the controlled agent achieves over a method that does not use opponent modelling. Compared to an ideal baseline method that has access to opponent trajectories during execution, we observed a relatively small decrease in performance of LIOM in our specific test environments. Further research on how such models could be used for non-stationary opponents would be of interest. In particular, we plan on investigating two scenarios; the first is multi-agent deep RL, where different agents are learning concurrently leading to non-stationarity in the environment, which prevents the agents from learning optimal policies [21, 32]. Secondly, we would like to explore notions of "safety" to handle opponents which aim to deceive and exploit the opponent model [39, 11, 12].

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

## A  PSEUDOCODE OF LIOM

Algorithm 1 shows the pseudocode of LIOM.

---

**Algorithm 1** Pseudocode of the LIOM algorithm

---

**for** $m = 1, ..., M$ episodes **do**
    Reset the hidden state of the encoder LSTM
    Sample $E$ opponent policies from $\Pi_{-1}$
    Create $E$ parallel environments and
    gather initial observations
    $a_{1,0}, r_{1,0}, \leftarrow$ zero vectors
    **for** $t = 1, ..., H$ **do**
        **for** every environment $e$ in $E$ **do**
            Get observations $o_{1,t}$ and $o_{-1,t}$
            Compute the mean $\mu(z_t) = q(z|o_{1,t}, a_{1,t-1}, r_{1,t-1})$
            Sample action $a_{1,t} \sim \pi(o_{1,t}, \mu(z_t))$
            Sample opponent action $a_{-1,t} \sim \pi_{-1}(o_{-1,t})$
            Perform the actions and get $o_{1,t+1}, r_{1,t}, d_{1,t}$
        **end for**
        **if** $t \mod$ update_frequency $= 0$ **then**
            Gather the sequences of all $E$ environments in a single batch $B$
            Perform a gradient step to minimise (8)
            Perform a gradient step to minimise (9)
        **end if**
    **end for**
**end for**

---

## B  EVALUATION ENVIRONMENTS

Figure 8 presents instances of the three multi-agent environment that were used for the experiments.

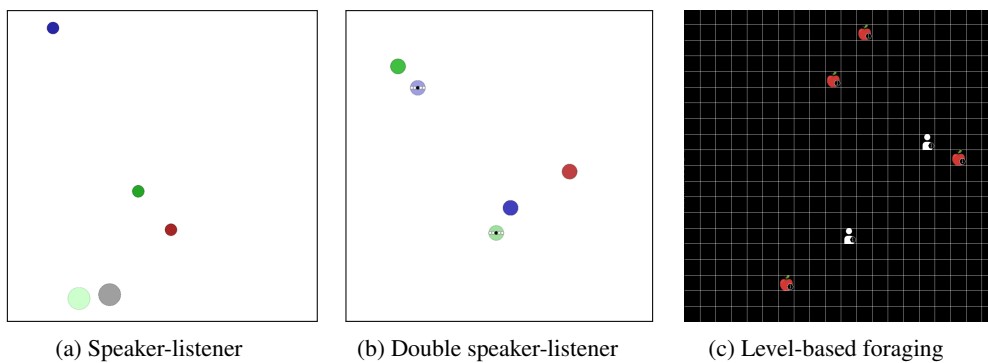

(a) Speaker-listener          (b) Double speaker-listener          (c) Level-based foraging

Figure 8: Multi-agent environments used in our evaluation.

## C  OPPONENT POLICIES

**Speaker-Listener:** Each policy of the speaker consists of a five-dimensional one-hot communication message that remains constant through the episode. We manually select different opponent policies that communicate different colours with different communication messages.

**Double Speaker-Listener:** Each of the opponent's policy consists of two sub-policies; one for the communication message and one for the navigation action. To generate diverse opponent, we fixed the communication action of all the ten agents to be the fixed policy that was used in speaker-listener. We then created five pairs of agents and we trained each pair to learn to navigate using the MADDPG

algorithm [28]. Each agent on the pair learns to navigate based on the communication message of the other agent in the pair.

**Level-Based Foraging:** The opponent policies in Level-based Foraging consist of 4 heuristic policies and 6 policies trained with MADDPG. The heuristic agents were selected to be as diverse as possible, while still being valid strategies. We used the strategies from Albrecht and Stone [1], which are: (i) going to the closest food, (ii) going to the food which is closest to the centre of visible players, (iii) going to the closest compatible food, and (iv) going to the food that is closest to all visible players such that the sum of their and the agent's level is sufficient to load it. We also trained 6 policies with MADDPG by training multiple pairs of agents and extracting the trained parameters of those agents. In order to circumvent the instability caused by deterministic policies in Level-based Foraging, we have found that enabling dropout in the policy layers [10] both during exploration and evaluation (thereby creating stochastic policies) the agents perform significantly better.

## D    IMPLEMENTATION DETAILS

All feed-forward neural networks have 2 hidden layers with ReLU [29] activation function. The encoder consists of one LSTM [37] and a linear layer with ReLU activation function. All hidden layers consist of 128 nodes. The latent dimension in speaker-listener and LBF is 15, while in double speaker-listener it is 20. The output of the decoder is passed through a softmax activation function to approximate the categorical opponent policy. For a continuous action space, a Gaussian distribution can be used. For the advantage computation, we use the Generalised Advantage Estimator [38] with $\lambda_{GAE} = 0.95$. We create 10 parallel environments to break the correlation between consecutive samples. The actor and the critic share all hidden layers. We use the Adam optimiser [25] with learning rates $3 \times 10^{-4}$ and $7 \times 10^{-4}$ for the A2C and the VAE loss respectively, and and we clip the gradient norm to 0.5. The multiplication factor of the MMD is $\lambda = 1$ for all the environments. We subtract the policy entropy from the actor loss [30] to ensure sufficient exploration. The entropy weight $\beta$ is $10^{-3}$. We train for 10 million steps in all environments. During the hyperparameter selection, we searched: (1) learning rates in the range $[10^{-4}, 7 \times 10^{-4}]$ and $[5 \times 10^{-4}, 10^{-3}]$ for the parameters of A2C and LIOM respectively, (2) hidden size between 64 and 128, and (3) entropy regularisation in the range of $[10^{-3}, 10^{-2}]$.

