# OpenReview forum: "Local Information Opponent Modelling Using Variational Autoencoders"
_ICLR.cc/2021/Conference — Reject_

### Official Review · AnonReviewer3 · 2020-10-23
**Official Blind Review #3**

**Rating:** 6
**Confidence:** 3

**Review:**

This paper considers an algorithm for learning and using an opponent model that is only conditioned on an agent's local information (history of actions, observations, and rewards). A variational autoencoder (VAE) is trained to predict opponent observations and actions, given the agent's local information. While the opponent's observations and actions are required to train the VAE decoder, they're not needed at play time for the encoder. The authors show that with static opponents, the output of the encoder is informative, and conditioning the agent policy on local information plus encoder output outperforms just using the local information.

The paper was well written, and was to easy follow. Experiments seemed appropriate to demonstrate the author's claims. I had only a few specific comments.

+++ Points after discussion about Meta-RL, HiP-MDPs, and MARL.

Like Reviewer 4, my view of MARL is more general than requiring all agents to be learning, even if most work does have one algorithm training multiple seats. I don't think this actually changes anything in the current draft, but should hopefully not appear in future edits.

The related works should include discussion of the HiP-MDP paper.

These points fit together: there should be a consistent placement of this paper, and related works. Given fixed opponents, the multiagent problem is equivalent to a single agent problem. It doesn't seem relevant whether or not the unobserved, unknown variables correspond to different but similar environments, or different but similar opponents in a single fixed environment.

+++

Section 4.2 "In Sections 1 and 2, it was noted that most agent modelling methods assume access to the opponent's observations and actions both during training and execution. To eliminate this assumption ..."
Weakened seems like a better choice of words than eliminated: it does still assume access to the opponent's observations during training.

Section 5
What is the size of the environment? Initial random placement? I'm trying to get some idea of the magnitudes of the rewards, which are based on Euclidean distance.
Are the speaker-listener and double speaker-listener experiments using the OpenAI multiagent particle environments? If so, cite this (The MADDPG paper that is already cited for opponent algorithms?)

---

> ### Author Response · Authors · 2020-11-16
> **Response to Reviewer 3**
>
> We would like to thank you for your review. We are delighted that you found our paper well-written and experiments appropriate.
>
> 1. We have updated the paper based on your recommendations to clarify that we eliminate this limitation only during execution.
>
> 2. We have updated the paper to include a more detailed description of the environments.

---

### Official Review · AnonReviewer1 · 2020-10-28
**A promising approach and a well-written paper; a few easily addressable concerns.**

**Rating:** 7
**Confidence:** 4

**Review:**

Summary:

This paper proposes an opponent modelling technique for imperfect information games.  During training, a VAE is trained to encode the agent's observed trajectory to a latent space, and then decode it back to the full trajectory (including opponent observations and actions).  The encoder can be used at runtime using only information the agent observes, and if the VAE latent means are included as an input to the agent's neural net (e.g., with A2C) then the agent's performance resulting performance is enhanced.  Empirical results are presented that support the technique's effectiveness.

----------

Positives:

- The paper is well motivated and clearly written.  I found it easy to read.

- The problem is significant and the approach is a natural fit.  Of course opponent observations should not be used at execution time, and training a VAE to recover the missing information seems like a good approach.  The empirical results support the technique and I found them convincing.

----------

Negatives:

- The paper is sometimes inconsistent in its description of techniques that need opponent observations at training time versus execution time.  The technique is often presented as "not needing opponent observations" while competing techniques "need opponent observations" (e.g., the end of Related Work:Learning Opponent Models, and the conclusion), whereas they mean (and often, but not always, clarify) that their technique *does* needs opponent observations during training, but does not during execution, and competing techniques need them during execution.  However, the authors appear unfamiliar with other opponent modeling work (I'm particularly familiar with the computer poker domain, although they cite two Ganzfried and Sandholm papers from that area) where there is a rich literature of opponent modelling being successfully used under these conditions (opponent observations needed only at training time, and never at execution).  The ongoing challenge in that community (with success against human professionals!) is to go further and perform opponent modelling without needing opponent obervations at *any* time, by using only the agent's observations at training time as well as execution time.  In general, the authors should be more careful about specifying "at execution time" whenever they claim their technique does not need opponent observations, and should probably be aware of more related work in this setting.

- "Opponent" modelling.  In the intro the authors describe using 'opponent' as a neutral term to mean 'other agents in the environment', and I'm sympathetic to that as I've encountered this awkward phrasing in my own work.  However, of the 3 environments described in this work, 2 are purely cooperative (Speaker-Listener, Double Speaker-Listener) and the third is a mixed setting involving cooperation and greedy behavior.  None of these three environments are a purely adversarial (i.e., zero-sum) setting where we would properly call the other player an 'opponent', or even a positive-sum or constant-sum setting that does not involve cooperation.  This is more important than just the choice of terminology, however!  In the experiments, I can imagine the LIOM technique could possibly be inaccurate but still perform well overall, since the other player is aligned with LIOM's goals and not rewarded for exploiting any mistakes.  In an *actual* adversarial setting, even if the opponent's policy was static (i.e., not updated in response to LIOM's policy, as in the cited Ganzfried papers), the opponent could still exploit any mistakes LIOM might make through its static policy, possibly driving its performance below that of NOM.  I liked the paper and I'm half joking when I say this, but: a better empirical analysis of an opponent modelling technique would involve at least one experiment that includes an *opponent* in the strict sense, to demonstrate that the technique is effective and robust in environments where true opponents exist.

- Empirical analysis.  I found this to be very good for supporting the main claims of the paper.  However, I would really have liked to see even a small investigation of robustness, even with static opponents (e.g., not with adapting opponents or worst-case opponents as in the Ganzfried and Sandholm papers cited in the conclusion, although I agree with the authors that that would be a great next step for future work).  For example, in Figure 2, if we were to generate additional holdout opponents for each game that were not part of the training set, and evaluate against them, how does the performance curve for LIOM and FIOM respond?  Do they still perform pretty well (e.g., they've usefully mapped the holdout opponent to something nearby in \pi_-1)?  Does it revert back to NOM?  Does it do *worse* than NOM, by making painfully incorrect assumptions about the opponent?  The answers here are pretty relevant to this work, since while we might train against a population of presumed opponents, our real opponents out in the world are unlikely to share their observations with us while we re-train!  If the technique remains a bit robust if we move outside it's training, then that's much more exciting than a technique that only works with its specific training set and fails dreadfully outside of it.  Further along these lines, I'd have loved to see a graph plotting 'Size of \pi_-1' on the x-axis and 'Average return after X steps' on the y-axis.  In such a graph, I'd really love to see: when '|\pi_-1| == 1' does NOM perform better than in Fig 2?  How quickly does LIOM (and FIOM?) drop off as we increase the opponent population size?  Is there something special about the population sizes used here, or can the approach scale well outside of those values?

- Related to the above two points: while the paper describes NOM as a lower-bound for LIOM, and it makes sense narratively to "bound" LIOM between FIOM and NOM, it really is not a lower bound in a mathematical sense.  FIOM might do worse than NOM in cases where 1) the other player is adversarial, or 2) the other player encountered at execution time has not been trained against, or 3) the other player was trained against, but insufficiently to create an accurate model, and overfitting to a bad model can be far worse than having no model.  In other opponent modelling work, I've seen opponent-aware agents do *far* worse than a baseline agent in those settings.

----------

Recommendation and Justification:

I think the paper should be accepted.  The topic is relevant, the proposed solution seems like a natural fit, and the empirical results are pretty extensive (measuring reward and prediction accuracy) and convincing in their support of the technique.

I have a few issues in the Issues and Suggestions section - some minor clarity points on the environments, a larger (but easily fixable) point about consistently describing when opponent observations are and aren't needed, suggestions for experiments I would have loved to see to verify basic robustness, and some notes about missed related work that also occupies this setting (use opponent info for training, never at execution).  Only one point (a line in the conclusion about this being the first successful use of opponent modelling under these conditions) is a strict concern; so long as that is addressed (or convincingly rebutted, if the authors disagree with the prior works I've cited), then I feel the paper is over the bar for acceptance.

I don't have specific questions for the authors, but if they'd like to accept, address, or rebut anything I've listed in Negatives or Issues and Suggestions, I'd certainly welcome their response.

----------

Issues and Suggestions


- Experiments section.  The description of the three games left me with quite a few questions (what exactly is observed by the players?  Are Speaker-Listener and Double-Speaker-Listener gridworlds, or something else? etc), and Appendix B provided helpful pictures but didn't clarify the fine details.  Since the Appendix is outside your page limit, can you be more explicit about exactly what the agents observe (pixels? If so, the whole map or a small region near them? If not pixels, how is the input formatted?  In Level-based Foraging, do agents observe their own level? The other player's level?) and what actions they take.  As a further example, for Speaker-Listener - the text says "each agent and landmark is randomly assigned a color".  Is the Speaker assigned a color?  It sounds like it from the text, but it wouldn't be useful for reward, so I'm guessing not?

- More on the game descriptions.  Speaker-Listener is the "Cooperative Communication" game from Lowe 2017, right?  If so, it seems appropriate to cite its source!  If not, best to note the differences.  Are the other games original to this work, or should they be cited also?

- Level Based Foraging.  It's not abbreviated as LBF while describing the game, but that abbreviation is used later, and that threw me for a bit.  I suggest abbreviating it in the description.

- Speaker-Listener and Double Speaker-Listener.  This is pretty minor, but I'm a bit concerned about a specific detail of these environments (especially S-L) and A2C that might give LIOM an advantage over NOM by leaking information, without having to model the opponent well at all (although from Fig3, Fig5, and Tab1, it appears to indeed model the opponent well).  Specifically, the VAE encoder is trained with the agent's (Obs, Reward, Act) stream, so it could encode something about Reward.  Does the agent's A2C component take in only Obs as an input, or does it also include Reward and Last Action?  The original A2C paper describes only taking Obs as input, but in the practical A2C implementations in the codebase I use, the last timestep's reward and last timestep's action are often used as inputs (concatenated on after the Obs convolutional layers, before the LSTM) as these "observations" often aid the agent's learning and asymptotic performance.  So here's the catch.  If the VAE encodes something about the last timestep reward, and the A2C component does not (i.e., NOM doesn't see it), then LIOM might have access to a valuable feature that NOM does not, unrelated to opponent modelling altogether.  And in S-L, I can imagine a Listener successfully solving the game even with a mute Speaker, just by taking steps and seeing what direction it has to go for its reward to go up.  Could an LSTM learn that?  Possibly!  I know Speaker-Listener is a standard game, but I'd be curious to see results with either 1) an A2C implementation that observes last_reward and last_timestep to see if NOM (or LIOM (Act,Rew)) improves, or 2) a sharp version of the environment where reward is granted on reaching the goal, just to confirm that LIOM isn't winning because it's being leaked more information than NOM can observe.

- In the conclusion, the statement "To the best of our knowledge, this is the first study showing that effective opponent modelling can be achieved without requiring access to opponent observations." is untrue (well, the "this is the first" part, not the "to the best of our knowledge part"  :^D  ).  The authors mean (and should clarify) "...at execution time", but even then, there is a rich literature of prior work.  I'm most familiar with the work in the computer poker domain, where opponent information is only used (if at all) at training time and never at execution time, some approaches do not need opponent information at either training or execution time, and the techniques have been successfully deployed in real poker games to defeat top professional human adversaries.  For example, in addition to the two Ganzfried and Sandholm papers cited in this work, consider:

 - Computing Robust Counter-Strategies, NeurIPS 2007, Johanson et al.  Uses opponent observations at training time to compute robust counter-strategies, uses agent observations at execution time to choose which counter-strategy to play against the current opponent (who may not be one of the opponents used at training time). The UCB method described in this paper was used to compete against human professional poker players in Heads-up Limit Texas Hold'em in the 2007 Man-vs-Machine Poker Championship, which the humans narrowly won.  In the paper's experiments against artificial opponents, opponent observations were used at training time but not execution time, and some experiments involved a holdout opponent for which no opponent observations were provided at any time.  In the competition against humans, no human data was used at training time or execution time, and only agent observations were used to decide how to adapt to the opponent.

 - Strategy Evaluation in Extensive Games with Importance Sampling, ICML 2008, Bowling et al.  The same conditions were used as above (use opponent observations for training robust counter-strategies, use only agent observations at runtime), but adds an unbiased, low-variance method for considering all possible states the opponent could be in given what the agent observed, and selecting a counter-strategy to use in response.  This method was used to defeat top human professionals for the first time in Heads-up Limit Texas Hold'em in the 2008 Man-vs-Machine Poker Championship.  In that event, no opponent observations were used at either training time or execution time: an even purer application than is described here, although I definitely agree that the strictest constraint must be to disallow opponent observations at runtime.

 - Online Implicit Agent Modelling, AAMAS 2013, Bard et al.  This technique is similar to the above, in that it does not use opponent observations at runtime at all, and can be used without any opponent observations at training time either.  This technique focuses on the creation of a portfolio of useful counter-strategies for use against a broad set of opponents, such that at least one counter-strategy will be effective against any particular (and previously unknown, not trained against) opponent. The "Implicit Opponent Model" approach is to model opponents by how we can respond to them, instead of trying to capture (unobserved!) information about how they act at individual decision points, and is the cleanest refinement of the approach used in the previous two papers.  Again, this is rhetorically purer than the approach described in this paper, in that it never needs access to opponent information at training or execution time.

These are just some of the works that I'm most familiar with; the computer poker community has been studying this topic for a while, and under the conditions described in this paper (opponent observations available during training, but never at execution time).  I'm not listing these citations as a suggestion to cite them (they are related to this work, and the topic is much broader and longer than the papers cited by the authors, but this paper is fine without the particular citations I've listed above).  I only list them as evidence to disprove the conclusion's assertion that "this paper is the first study showing that effective opponent modelling can be achieved without access to opponent observations (at execution time)", as that is definitely untrue.

----------------------

I've read the other reviews, the author's responses, and the discussion - thank you everyone.  I'm still in favour of accepting the paper.

---

> ### Author Response · Authors · 2020-11-16
> **Response to Reviewer 1**
>
> We would like to thank you for your extremely thoughtful and detailed review - without a doubt one of the best reviews we have seen.  We are delighted that you found the paper well-written and the results convincing. We address specific comments below:
>
> “Negatives:”
> 1. We have updated the paper to be consistent throughout about our assumptions (clarifying distinction between training and execution time). We also included some of the mentioned Poker works in Section 2 and discuss the relation to our work. Thank you for pointing us to these papers.
>
> 2. We understand your concern regarding the terminology. Our choice to use the term “opponent” was purely out of convenience, as we believe that this terminology makes it easier to distinguish between the controlled agent and the other agent in the environment. We agree regarding the comment on exploitability, and we have left this question to future work. It would be interesting to evaluate how and to what extent a method such as LIOM could be exploited by an adversary.
>
> 3. Following your recommendation, we added Section 5.6 in the updated paper, in which we evaluate the effect of the number of opponent policies. We agree that further analysis of robustness against opponent policies out of the training set \Pi_{-1} is an important issue and it is left as future research.
>
> 4. We have removed the term “bound”, based on your recommendation.
>
> “Issues and Suggestions:”
>
> 1,2,3.  We have added clarifications to the paper based on your comments.
>
> 4. In the speaker-listener environment the reward does not leak because we do not back-propagate the gradient from the RL loss to the parameters of LIOM. However, from Figure 3, we observe that the reward is necessary for generating efficient opponent models. If the reward from the last time step is used with a fully-connected network then the return of NOM does not change. If the reward from the last time step is used with an LSTM then the agent learns to follow the direction that decreases the reward and results in an average return of -10. When the reward is used in the opponent model, in LIOM, it can significantly improve the returns, which are close to -6. In the double speaker-listener there is no difference in the returns of NOM in any of the aforementioned cases.
>
>
> 5.  We have updated the paper based on your recommendation. We have also included a paragraph in the related works section to cite the proposed references.  The main difference between our work from the references that you provided is that they have multiple episodes to adapt to a single opponent. For example, Johanjon et. al. 2007 use UCB1  to select the best policy against a specific opponent for 1000 episodes. In contrast, our method uses a single episode to infer the opponent model.

---

> > ### Comment · AnonReviewer1 · 2020-11-17
> > **Minor detail - 1000 episodes**
> >
> > Just clarifying the poker papers 'multiple episodes to adapt' bit.  There's not a clean comparison between episodes/timesteps in poker and RL problems, since many poker "episodes" will be just 1 or even 0 timesteps long from one agent's perspective (if you fold, or if your opponent folds and the game is over before your first turn) and even the longest "episodes" are only 10-15 timesteps long.  So it's a better comparison in a repeated game for one episode to consist of many iterations of the stage game, which themselves consist of timesteps, making a 1000-hand poker match more like one episode of 5k to 10k timesteps of interaction, instead of 1k episodes.
> >
> > Putting that detail aside though and treating individual poker hands as episodes...  In all of the linked papers, the agents performed adaptation after each episode, and continued to adapt over the course of the match.  For example, in that UCB1 experiment from Johanson 2007, the agent adapted after every episode (which was only 0-15 timesteps long), using the history of interaction that had happened over all episodes so far, and continued to adapt over the match of 1000 episodes (~10k timesteps).  It didn't need 1000 episodes of historical data to pick a policy to use, and wasn't choosing a policy to use for the next 1000 episodes.  So the speed of adaptation is quite a bit more similar to this setting than your comment suggests.

---

> > > ### Author Response · Authors · 2020-11-17
> > > **Thank you for the clarification**
> > >
> > > Thank you for your clarification. This was what we originally intended to express. We will also attempt to clarify this in our text if the paper gets accepted.

---

### Official Review · AnonReviewer4 · 2020-10-28
**I tend to reject this paper because of its lack of novelty, the proposed idea has been extensively used in centralised training, decentralised execution multi-agent (Dec-POMDP) settings.**

**Rating:** 3
**Confidence:** 4

**Review:**

The authors of this paper study opponent modelling in partially observable Markov games, following the _centralised training, decentralised execution_ paradigm. In particular, during training the assume access to all agents' (both ego and other agents) local observations and actions, while at test/execution time they control the ego agent with only local information. The local ego information, i.e., ego local observations, actions and rewards are used to infer unobserved (sufficient) statistics for others' observations and actions. The authors show empirically that their method perform competitively in a set of standard multi-agent environments.

**Questions and Concerns**:
1. The claims about novelty and comparison to literature is questionable. The authors **do** use non-local information during trained to train their VAE, an idea that is very much common in the "centralised training, decentralised execution multi-agent paradigm, i.e., Dec-POMDPs".
2. How does the proposed method do compared to standard Dec-POMPD solvers, e.g., [1-4]?

**References**

[1] Foerster, J., Farquhar, G., Afouras, T., Nardelli, N. and Whiteson, S., 2017. Counterfactual multi-agent policy gradients. arXiv preprint arXiv:1705.08926.

[2] Rashid, T., Samvelyan, M., De Witt, C.S., Farquhar, G., Foerster, J. and Whiteson, S., 2018. QMIX: Monotonic value function factorisation for deep multi-agent reinforcement learning. arXiv preprint arXiv:1803.11485.

[3] Sunehag, P., Lever, G., Gruslys, A., Czarnecki, W.M., Zambaldi, V., Jaderberg, M., Lanctot, M., Sonnerat, N., Leibo, J.Z., Tuyls, K. and Graepel, T., 2017. Value-decomposition networks for cooperative multi-agent learning. arXiv preprint arXiv:1706.05296.

[4] Jaques, N., Lazaridou, A., Hughes, E., Gulcehre, C., Ortega, P., Strouse, D.J., Leibo, J.Z. and De Freitas, N., 2019, May. Social influence as intrinsic motivation for multi-agent deep reinforcement learning. In International Conference on Machine Learning (pp. 3040-3049). PMLR.

---

> ### Author Response · Authors · 2020-11-16
> **Response to Reviewer 4**
>
> We would like to thank you for your review.
> Please note that the provided references focus on multi-agent reinforcement learning (MARL), which is a different problem from our setting. In MARL, we control multiple agents and the goal is to train optimal coordinated policies for the agents to solve a given task. In contrast, our work focuses on a setting in which we control a single agent while the other agents draw their policies from a set of defined (fixed) opponent policies. We make this explicit in Section 1, fourth paragraph (“... other agents do not simultaneously learn, such as in multi-agent RL”) and state our setting formally in Section 4.1 (problem formulation). We have added one paragraph in Section 2, to further clarify how the problem explored in our paper is different from the one that MARL tries to address.
> Thus the MARL methods in the papers given by the reviewer are not directly applicable to our problem setting. We hope that based on our response the reviewer will reconsider their review accordingly.

---

> > ### Comment · AnonReviewer4 · 2020-11-22
> > **Response to Authors' Rebuttal**
> >
> > Thank you for your reply. Maybe to rephrase my concerns with the novelty of your work:
> >
> > > In contrast, our work focuses on a setting in which we control a single agent while the other agents draw their policies from a set of defined (fixed) opponent policies.
> >
> > The problem setting you have selected is such that at training time you are given access to privileged (i.e., non-local) information about other agents (i.e., their observations and actions). At test time, you only access local information (i.e., ego observations, actions and rewards). This is a data imputation problem, i.e., at test time you have missing values which you try to infer using observed data. The fact that others' policies are fixed and drawn from a possible finite set make the problem **stationary** and hence it's equivalent to a HiP-MDP [1] and your LIOM is nothing else than the standard filtering approach to HiP-MDPs.
> >
> > > Please note that the provided references focus on multi-agent reinforcement learning (MARL), which is a different problem from our setting. In MARL, we control multiple agents and the goal is to train optimal coordinated policies for the agents to solve a given task.
> >
> > I would like to disagree with your definition of MARL. There is nothing that forces **all** agents to be learners. It's perfectly valid to have fixed agents in a MARL problem setting. MARL is a general problem setting that deals even with the scenario that others' learn too, a more general framework than the one you address here.
> >
> > Overall, I decrease my original evaluation since it is now clear to me that you solve a HiP-MDP problem where at training time you have access to the Markov state (a rather strong and less general assumption than standard HiP-MDP solvers) and only at test time you partially observe the state and infer the hidden parameters (i.e., sufficient statistics for others' policies) with a standard data imputation method based on VAEs.
> >
> > **References**
> >
> > [1] Doshi-Velez F, Konidaris G. Hidden parameter Markov decision processes: A semiparametric regression approach for discovering latent task parametrizations. In IJCAI: proceedings of the conference 2016 Jul (Vol. 2016, p. 1432). NIH Public Access.

---

> > > ### Author Response · Authors · 2020-11-23
> > > **Response to Reviewer 4**
> > >
> > > Thank you for pointing us to this work which we were not aware of. This is a latent variable model similar to many that exist in the literature, and we will gladly add a reference to this paper. Indeed we have cited several related works (see “Representation Learning in Reinforcement Learning”) that solve closely related problems by trying to learn latent factors over the transition and reward functions. Note that there are significant differences between the HiP-MDP work and our work: they assume some single-agent tasks while we assume multi-agent interaction tasks; and they use Indian Buffet and Gaussian Processes (IBP-GP model) for inference while we use VAEs with an encoder which works on locally observed information. Please note that we do not assume access to the state of the environment (as incorrectly claimed by reviewer), only to local and opponent trajectories during training. Indeed, opponent modelling in multi-agent systems under such limited observability was described as a significant open problem in a recent survey [1] and our work aims to address this problem.
> > >
> > > > I would like to disagree with your definition of MARL. There is nothing that forces all agents to be learners. It's perfectly valid to have fixed agents in a MARL problem setting. MARL is a general problem setting that deals even with the scenario that others' learn too, a more general framework than the one you address here.
> > >
> > > The common assumption in MARL papers (including all references provided by the reviewer) is that the learning algorithm learns policies for multiple agents. The cited MARL algorithms are not designed for our problem setting, in which we control a single agent which must learn to interact with other agents whose (unknown) policies are drawn from a set of policies.
> > >
> > > [1] Stefano Albrecht, Peter Stone. "Autonomous agents modelling other agents: A comprehensive survey and open problems." Artificial Intelligence 2018

---

> ### Author Response · Authors · 2020-11-22
> **A respectful request**
>
> As we are nearing the end of the rebuttal phase, we would be grateful if the reviewer could respond to our comments and reconsider their review in light of our clarifications.

---

### Official Review · AnonReviewer2 · 2020-10-29
**Well-executed study of learning opponent models with a VAE**

**Rating:** 6
**Confidence:** 3

**Review:**

This paper addresses interactions in a multi-agent setting without access to the policies of other agents (termed opponents). Each agent has access to local observations only but is still required to cooperate with opponents (which use either heuristic or stochastic RL-trained policies). In each instance of the environment, the agent considered interacts with single opponent, drawn from a fixed set. The idea is then to learn an embedding of each opponent, which is derived from local observations of the agent, and trained by predicting opponent observations, rewards and actions which are made available at training time.

Overall, I find this to be a well-executed study of learning opponent models with a VAE. The write-up is clear and easy to follow, and the experiments show good performance of the method, along with a base- and topline. Ablations on VAE training targets are provided, as well as analysis regarding the learned opponent embeddings and VAE decoder performance. I'm however a bit surprised by the poor performance of the NOM baseline on the speaker-listener environment. The reward is dense, so it seems the main task for the listener (mapping the the available 5-bit message to the respective goal location) should be solvable as-is?

Some suggestions for further improvement:
- I found Figure 1 confusing as I understand that the actual opponent policies do not have access to the latent variable Z, and that three items are predicted from the latent space: the opponent's actions, observations and rewards.
- For LBF in Figure 3, I would suggest that the authors expand upon the fact that there is no discernible structure in the opponent embeddings. It would have been interesting to see how well the method works if agent and food locations were randomized in the LBF environment so that agents also have to cooperate in exploration.
- I would also suggest to clearly state the dimensionality of the learned embeddings in Appendix D -- from the current text I would assume it is 128?
- In the conclusion, you state: "LIOM is agnostic to the type of interactions in the environment (cooperative, competitive, mixed) and can model an arbitrary number of opponents simultaneously." The method presented appears to only support a single opponent per environment instance, so it would be good to clarify this statement.

---

> ### Author Response · Authors · 2020-11-16
> **Response to Reviewer 2**
>
> We would like to thank you for your review. We are delighted that you found the paper well-written and well-executed.
>
> Regarding the speaker-listener environment, the NOM agent cannot separate different opponent policies, and learns an “average” policy that learns to move into the centroid of the 3 landmarks.
>
> 1. The opponent does not have any access to the VAE neither during training nor execution.
> Thank you for bringing that to our attention. We updated the text from the two boxes in the decoder of Figure 1 to make it clear:
> “Opponent policy” -------> “opponent policy reconstruction”
> “Opponent observation” -------> “opponent observation reconstruction”
>
> 2. Regarding LBF, the agents’ and foods’ positions are randomly generated at the beginning of the episode. We have updated the LBF description accordingly to make it clear.
> Regarding the embeddings: We use PCA to project the embeddings, and as a result separate clusters are not always visible as in the previous 2 environments. As we mention in section 5.3 the embeddings are generated to capture the opponent’s actions and private observations. In Figure 3, we plot embeddings for 200 different episodes, for each opponent policy, from the 20th time step of the episode. Because of the random initial state, the same opponent policy has a different private observation and performs a different action at the 20th time step, between different episodes, resulting in an embedding space where different clusters are not clearly visible.
>
> 3. We have updated the paper’s appendix based on your recommendation.
>
> 4. Regarding the sentence in the conclusion, we refer to the number of opponent policies in the set \Pi_{-1} and, not to the number of the opponents in the environment.  We have updated the paper based on your recommendation to make it clear.

---

### Author Response · Authors · 2020-11-16
**General Response**

We would like to thank all the reviewers for their time and their valuable feedback. We have uploaded a revised version of our paper that addresses some of the issues that were mentioned by the reviewers.  The main differences are:

(i) section 5.6, where we present the effect of the number of opponent policies in the returns of FIOM, LIOM, and NOM. Please note that one seed is used for generating Figure 7. We will update the figure with the results from five seeds if the paper gets accepted.

(ii) two new paragraphs in section 2. The first one discusses opponent modelling in Poker research, and the second the differences between our work and multi-agent reinforcement learning.

---

### Decision · Program_Chairs · 2021-01-07
**Final Decision**

**Decision:**

Reject

**Comment:**

The submitted paper is well written and easy to follow and also the idea of using VAEs for making inferences about the opponents on which a policy can be conditioned on is sensible. Also the reported performance in comparison to two baselines is good (although I have concerns about the selection of the baselines—see below). Acceptance of the paper was suggested by 3 of the reviewers and rejection by one of the reviewers. While I don’t share all concerns of the negative reviewer, I also suggest to reject the paper.

My suggestion to reject the paper is mainly based on seeing concerns of the positive reviewers more critical as these reviewers themselves and some concerns I have on my own. In particular, I don’t think that all MARL approaches can simply be discarded for comparison—no matter whether the opponents are learning or not. Regarding the evaluation, I think that an environment with real opponents must be considered and that robustness is a key property that should be studied (otherwise an approach with a fixed set of best response policies and inference about the opponent might perform as well). In that regard I also find the selection of baselines insufficient—the minimum I would expect is to consider a NOM baseline using an RNN (which as far as I can tell is not the case) such that it could make inferences about the opponent.

I want to acknowledge that the paper improved quite a lot during the rebuttal period in which the authors extended their discussion of related work on opponent modeling.

In summary, the paper could be improved substantially by an extended empirically study (more environments + baselines + "mismatch" settings). If the currently observed performance gains also hold in these settings, this can become a good paper but in its current form I think the paper is not demonstrating that the proposed approach performs favorably over natural baselines and works well against real opponents.